# Treatment of Pathologic Peritrochanteric Fractures Using Sliding Hip Screws Augmented with Cerclage Reconstruction Plates

**DOI:** 10.3390/jcm10184271

**Published:** 2021-09-21

**Authors:** Ying-Kuei Kuo, Hsuan-Yu Chen, Yuan-Fuu Lee, Ting-Chun Huang, Tsung-Han Yang, Yu-An Chen, Rong-Sen Yang

**Affiliations:** 1Department of Orthopedics, Taipei Hospital, Ministry of Health and Welfare, New Taipei City 242, Taiwan; daniel75516@gmail.com; 2Department of Orthopedics, National Taiwan University College of Medicine, National Taiwan University Hospital, Taipei 100, Taiwan; hychen83@gmail.com; 3Department of Orthopedics, National Taiwan University Hospital Hsin-Chu Branch, Hsinchu 300, Taiwan; G03808@hch.gov.tw (T.-C.H.); johnyang19891101@hotmail.com (T.-H.Y.); 4Department of Orthopedics, Lo-Sheng Hospital, Ministry of Health and Welfare, New Taipei City 242, Taiwan; b95401086@ntu.edu.tw; 5College of Medicine, National Taiwan University, Taipei 100, Taiwan; alexchen996@gmail.com

**Keywords:** cerclage reconstruction plate, augmentation, sliding hip screw, peritrochanteric, pathologic fracture, impending, metastasis

## Abstract

We proposed a new method to augment the traditional sliding hip screw (SHS) with cerclage reconstruction plates to treat pathologically impending and actual peritrochanteric fractures as well as to revise open reductions and internal fixations to increase the construct strength against the shearing force, thus reducing the implant failure rate. In this retrospective study, patients with peritrochanteric pathology with at least two years of follow-up who underwent augmentation with cerclage reconstruction plates (modified SHS) and conventional SHS between 1 May 2015 and 31 May 2017 were divided into groups A (*n* = 12) and B (*n* = 28), respectively. Demographic data, surgery duration, blood loss, complications, and local radiotherapy were analyzed. The average surgery duration was significantly longer in group A (*p* = 0.013). The estimated intraoperative and perioperative blood losses were not significantly different between the groups. The implant survival rates were not significantly different under competing risk analysis. The success rate of a revision surgery with modified SHS was excellent, and implant survival time was >2 years, as observed with the previous SHS constructs. Subtrochanteric region involvement and a postoperative visual analog scale ≥4 could be risk factors of implant failure and revision surgery. This technique can be an alternative treatment for difficult pathologic peritrochanteric fractures, especially those with previous plating failure.

## 1. Introduction

The bone is one of the most common sites of metastasis in patients with cancer, and approximately 56% of cases involve the long bones in the lower limbs [1]. Pathologic fractures are common in patients with bone metastases (9–29%) [2]. Given the stress concentration in the peritrochanteric femoral region, 51% of pathologic fractures were reported to have occurred in the femur, and the peritrochanteric region is the site for 25.9% of fractures [3]. Pathologic impending or actual peritrochanteric fractures often require forced immobilization due to the severe pain associated with it [4], and surgical procedure is vital to maintaining patient function and quality of life. Factors such as tumor location, tumor type, cortical destruction, life expectancy, and financial issues need to be considered in making decisions regarding the treatment strategy.

Trends in the surgical treatment of pathologic peritrochanteric fractures have been moving toward intramedullary nailing (IMN) and endoprosthetic reconstruction (EPR) [5]. According to an online survey that was undertaken by members of the Musculoskeletal Tumor Society in 2012, IMN was performed in 45% of the patients, proximal femur resection and reconstruction were performed in 34%, long-stem cemented hemiarthroplasty and cemented hemiarthroplasty were performed in 15%, and open reduction and internal fixation (ORIF) were performed in 7%. However, the major concerns associated with IMN stabilization for femoral metastasis include tumor recurrence or spread, deep venous thrombosis, hard to revise in an implant fracture, and infection [6]. Moreover, reaming, which is necessary in IMNs, may cause tumor seeding to the distal end of the femur [7].

Sliding hip screws (SHS) have been used to treat pathologically impending or actual peritrochanteric fractures for decades, and they may provide a better field for tumor debulking and curettage, easier fracture reduction, cement augmentation, limiting the surgical site to the proximal femur region, and cases contraindicated by IMN. Moreover, revision surgery is more difficult in IMN, especially when further revision to EPR is indicated. Additionally, the amount of host bone that requires resection is less when the previous surgery was performed using the SHS construct [6].

Nonetheless, there are drawbacks in using conventional SHS in the treatment of pathologically impending or actual peritrochanteric fractures. The complication rate with the SHS construct is reportedly as high as 42%, compared with that of EPR (3.1%) and IMN (6.1%); however, SHS remains the necessary choice for surgical treatment of pathologic peritrochanteric fractures [8]. Figure 1 shows a difficult case with multiple implant failures treated using conventional SHS technique complicated with cement augmentation loosening, non-union and broken screws, and all these increased difficulty in revision surgery. Longer SHS side plates are needed to provide better strength against the shearing force in each revision surgery. Therefore, in this retrospective comparative study, we compared two methods: a modified SHS technique augmented using cerclage reconstruction plates to reduce implant failure rate and to provide an alternative when performing revision ORIFs to treat patients with pathologically impending or actual peritrochanteric fractures caused by bone metastases, and conventional SHS. We also aimed to identify the factors that are associated with revision surgery.

## 2. Material and Methods

### 2.1. Study Design

In this retrospective study, we included patients with pathologically impending and actual peritrochanteric fractures caused by bone metastases who underwent fracture fixation using SHS by a single surgeon, the senior author, i.e., Rong-Sen Yang, Chief of the Department of Orthopedic Surgery, National Taiwan University Hospital, Taiwan, between 1 May 2015 and 31 May 2017. All lesions were solitary, and no skip lesion was observed in the other parts of the same femur. In this study, the medical history of each patient was reviewed, and all patients underwent the following: physical examinations, blood tests (tumor markers), plain radiography, regional MRI, computed tomography (CT of the chest, abdomen, and pelvis), whole-body technetium-99m (Tc-99m) bone scintigraphy, and biopsy. The indications for SHS fracture fixation were impending and actual peritrochanteric fractures with sufficient bone stock, life expectancy >3 months based on the Tomita score, and high fracture risk based on the Mirels score. Patients were excluded if (1) they had a pathology other than a metastatic tumor, such as a primary tumor, infection, tuberculosis, or traumatic injury; (2) Other surgical techniques, such as IMN or EPR fracture fixation; (3) a follow-up period <2 years; or (4) incomplete data.

Patients were divided into two groups: Group A comprised patients who underwent the modified SHS technique with augmentation using cerclage reconstruction plates in addition to curettage, cementing, and internal fixation with SHS. Group B comprised patients who underwent curettage, cementing, and internal fixation with SHS only, without cerclage reconstruction plates. Figure 2 shows the flowchart of the study. 

### 2.2. Surgical Procedure

Patients were intravenously administered 1000 mg of tranexamic acid (TXA) preoperatively. All procedures were performed with the patient in a lateral decubitus position, and the lateral approach was used to split the tensor fascia. An L-shaped incision was made at the proximal origin of the vastus lateralis muscle; thereafter, the muscle was detached and rotated anteriorly. Careful dissection with electrocauterization was performed, and perforating arteries were ligated as needed. A lateral bone window was created with the lag screw reamer if it was an impending fracture; if the bone had already been fractured, we utilized the fracture site to perform intralesional or marginal curettage of the bone tumor. The surgical area was routinely irrigated with 95% alcohol solution to devitalize the residual invisible tumor cells, but irrigation with normal saline was also performed to prevent subsequent burning of the alcohol during electrocauterization. Standard guided pin insertion and reaming were performed under C-arm fluoroscopy. A disposable sputum suction tube was placed in the medullary canal to create a negative pressure; subsequently, vancomycin-impregnated polymethylmethacrylate (PMMA) cement was tamped into the bone defect, including the femoral shaft and femoral head. Before the cement solidified, the lag screw was inserted under C-arm fluoroscopy guidance. Fracture reduction was maintained. Once the cement hardened, the cortical screws for the side plate were inserted.

In cases where the cerclage augmentation technique was performed, a 3.5 mm AO reconstruction plate was bent into a C-shape after consideration of the curve and length, and the reconstruction plate was then passed around the SHS side plate and femoral shaft (Figure 3a). The ideal position of the reconstruction plate wrapping around would be the two most distal screw holes of the SHS side plate. One cortical screw was initially passed through the screw hole of the reconstruction plate and the SHS side plate to maintain the position of the reconstruction plate, and a bone holder was used to tighten and create tension on the reconstruction plate while the other compression screw was inserted (Figure 3b).

After irrigating the wound with normal saline, the vastus lateralis muscle flap was reattached. A single 1/8-inch Hemovac drain was inserted into the wound, and the tensor fascia and wound were repaired in layers using continuous Vicryl sutures. Subsequently, 1000 mg of TXA was routinely injected via the Hemovac drain into the wound, and the tube was clamped for 4 hours before negative-pressure drainage was initiated.

### 2.3. Measurements

Demographic data, sex, body weight (BW), patient survival time, surgery duration, preoperative lesion status (impending or actual fracture), tumor type, pre- and postoperative visual analog scale (VAS), complications, and local radiotherapy were derived from the medical records. Radiographs were reviewed to assess the location of the lesion. The surgery duration is expressed in minutes. The following complications were recorded: poor wound healing, wound bullae formation, local tumor progression or recurrence, and implant breakage causing re-fracture or collapse requiring revision surgery. The first plain radiograph showing implant fracture was used to indicate the end of implant survival. The latest visit to the outpatient clinic was regarded as the end of the follow-up. If any patient’s latest follow-up was at least 6 months earlier than the date of the analysis, it was considered to have expired.

The estimated intraoperative blood loss (in mL) was recorded by subtracting the total volume of fluid irrigated from the total volume of fluid collected in the suction bottles. If the calculated volume was <50 mL, it was recorded as 50 mL. Preoperative hemoglobin (Hb_i_, g/dL), preoperative hematocrit (Hct_i_), units of transfused packed red blood cells (RBC), postoperative day 1 hemoglobin (Hb_e_), postoperative day 1 hematocrit (Hct_e_), BW (in kg), and body height (BH, in cm) were also recorded. Two methods were employed to calculate the overall perioperative blood loss: modified Meunier’s formula and modified Mercuriali’s formula. The blood volume was estimated using Nadler’s formula.

*Nadler’s formula* [9]:For men:
$$\mathrm{Estimated}\;\mathrm{blood}\;\mathrm{volume}=604+0.0003668\times {\mathrm{BH}}^{3}+32.2\times \mathrm{BW}$$For women:
$$\mathrm{Estimated}\;\mathrm{blood}\;\mathrm{volume}=183+0.000356\times {\mathrm{BH}}^{3}+33\times \mathrm{BW}$$*Meunier’s formula* [10]:
$$\mathrm{Estimated}\;\mathrm{blood}\;\mathrm{loss}=\mathrm{blood}\;\mathrm{volume}\times \frac{{\mathrm{Hb}}_{i}\;–{\;\mathrm{Hb}}_{e}}{{\mathrm{Hb}}_{e}}$$
*Our modification of Meunier’s formula:*

$$\mathrm{Estimated}\;\mathrm{blood}\;\mathrm{loss}\;\left(\mathrm{in}\;\mathrm{mL}\;\mathrm{of}\;\mathrm{blood}\right)\;=\mathrm{blood}\;\mathrm{volume}\times \frac{{\mathrm{Hb}}_{i}\;–{\;\mathrm{Hb}}_{e}}{{\mathrm{Hb}}_{e}}+250\times \mathrm{units}\;\mathrm{of}\;\mathrm{transfused}\;\mathrm{pRBC}$$

(In Taiwan, 1 unit of pRBC is processed from 250 mL of whole blood).*Original Mercuriali formula* [11]:
$$\mathrm{Estimated}\;\mathrm{blood}\;\mathrm{loss}\;\left(\mathrm{in}\;\mathrm{mL}\;\mathrm{of}\;\mathrm{RBC}\right)\;=\mathrm{blood}\;\mathrm{volume}\times \;\left({\mathrm{Hct}}_{\mathrm{preop}}-{\mathrm{Hct}}_{\mathrm{day}\;5\;\mathrm{postop}}\right)\;+\mathrm{mL}\;\mathrm{of}\;\mathrm{transfused}\;\mathrm{RBC}$$
*Our modification of Mercuriali formula:*

$$\mathrm{Estimated}\;\mathrm{blood}\;\mathrm{loss}\;\left(\mathrm{in}\;\mathrm{mL}\;\mathrm{of}\;\mathrm{RBC}\right)\;=\mathrm{blood}\;\mathrm{volume}\times \;\left({\mathrm{Hct}}_{i}-{\mathrm{Hct}}_{e}\right)\;+100\times \mathrm{units}\;\mathrm{of}\;\mathrm{transfused}\;\mathrm{RBC}$$

(In Taiwan, 1 unit of pRBC contains ~100 mL of RBCs)

### 2.4. Statistics

The Mann–Whitney U test was used to compare two independent groups of nonparametric data. The Shapiro–Wilk test was used to test normality. Multivariate Cox regression analysis was used to evaluate the factors associated with the endpoints of implant failure and revision surgery in all patients. A competing risk analysis was used to express implant survival under the consideration of death in groups A and B. All *p*-values were two-tailed, and a *p*-value < 0.05 was considered significant. All statistical analyses were performed using MedCalc Statistical Software version 18.5 (MedCalc Software bvba, Ostend, Belgium; 2018).

## 3. Results

Group A included 12 patients, and group B included 28 patients; all were followed up for at least 2 years. The patient demographic data included age, sex, BW, preoperative lesion status (impending or actual fracture), tumor location, and tumor type (Table 1).

No significant differences were found in the patient demographics and in the overall survival between groups A and B. Group A had a significantly longer surgery duration and higher rates of actual fractures. Metastatic lung cancer was the most common tumor type in both groups. The estimated perioperative blood loss calculated using Meunier’s formula and Mercuriali’s formula showed no significant difference between the two groups. The complications observed in both groups are listed in Table 2.

Two implant failures in group A required revision surgery; both patients underwent EPR. Five implant failures in group B required revision surgery, which was performed using the modified SHS technique (Figure 4). The success rate of revision surgery was 100% (5/5) without using a longer SHS side plate, and the implant survival time was >2 years than that observed with the previous SHS constructs.

After controlling for confounding factors, tumor location (presence or absence of subtrochanteric region involvement) and postoperative VAS score were identified as independent factors associated with a higher risk of implant failure and revision surgery. The cause-specific hazard risk associated with tumor location involving the subtrochanteric region was 2.95 (95% CI, 1.23–6.75; *p* < 0.0001), and the postoperative VAS score ≥4 was 2.19 (95% CI, 1.35–3.57; *p* < 0.002) (Table 3).

The implant survival rate at a given time point was calculated by dividing the number of surviving implants with the total number of surviving patients at a given time. The competing risk analysis showed no significant difference in the survival of implants between groups A and B; however, a superior trend for implant survival was observed in group A (Figure 5).

## 4. Discussion

Patients with pathologically impending or actual peritrochanteric fractures usually have a history of a malignant condition, and accurate diagnosis of the fracture is crucial before proceeding with surgical treatment. A successful diagnostic strategy to identify the primary malignancy in patients with skeletal metastases of unknown origin includes reviewing the medical history and performing physical examinations, laboratory analyses, plain radiography, Tc-99m bone scintigraphy, and CT of the chest, abdomen, and pelvis. Obtaining preoperative or intraoperative biopsies should also be considered to confirm the diagnosis. Tumor location, tumor type, extent of destruction, and general condition of the patient are the bases to decide further treatment. Since the survival rate in patients with cancer is improving owing to advancements in adjuvant therapies, implant failure may not occur with disease progression. Instead, implant failure may occur because the patient remained active and ambulatory. Improved patient survival has been associated with complications in surgery, and our study aimed to identify treatment-specific factors to patient-specific factors. The results of our study demonstrated that tumor recurrence and progression occurred at the previous lesion sites only under SHS technique, and none of the patients with tumor recurrence or progression experienced implant failure. After controlling for confounding factors, involvement of the subtrochanteric region and a postoperative VAS score ≥ 4 were identified as independent factors associated with a higher risk of implant failure and revision surgery. When tumors involve the subtrochanteric region, IMN, EPR, or a locking plate may be better choices than SHS. If the patient has a postoperative VAS score ≥ 4, the surgeon should be more watchful of any signs of implant failure.

Limb salvage surgery continued to be the main goal in treating patients with pathologic peritrochanteric lesion, and the choice of surgical treatment required careful individual considerations, including tumor staging, pathologic fracture status, and clinical condition. The extent of osteolytic destruction of the proximal femur and hip joint is critical to the choice of IMN, EPR, or SHS. Greater instability at the lesion site, vascular hyperplasia, tissue erosion, congestion and edema, and poor healing potential were expected in pathologically impending or actual peritrochanteric fractures. The use of stable fixation techniques such as IMN, EPR, and SHS, to provide sufficient stability that allows for immediate full weight bearing, with adequate local tumor control, and to extend life expectancy should be the treatment principles for these patients.

A previous study demonstrated a high frequency of re-revisions once a revision surgery has been performed [12], and surgeons should always be mindful of the revision strategy. Minimally invasive techniques combined with bone cement reinforcement were usually performed with IMN, but the complications of IMN implant failure still required further revision strategies. Larger diameters and longer IMNs were needed for revision surgery, but a history of using augmented cement or broken screws frequently increased the difficulty in revision IMN. In most cases of revision IMN, patients received custom-made total femur EPRs as a revision solution, but this was associated with potential poor prognosis overall. Performing revision ORIFs with SHS several times was possible by changing the direction of cortical screw insertion. Only a proximal femur EPR was needed if revision ORIFs failed at the very end, which has a much better prognosis than total femur EPR. EPR could be a good choice for patients with large pathologic osteolytic destruction, but local tumor recurrence, infection, aseptic loosening, prothesis breakage, recurrent dislocation, or low functional results were the main potential complications of EPR. EPR required more extensive revision surgery once complications occurred, so EPR was reserved for those who are not suitable to receive other surgeries or have experienced failure from internal fixation in our surgical strategy.

At our institute, SHS comprises 30–40% of surgical cases, since the open approach in the SHS technique provides good exposure for tumor resection, easier fracture reduction, and easier PMMA cementing; it is also economically convenient. Moreover, the muscle cuff around the joint is well preserved in the SHS technique compared with that observed in arthroplasty [13]. Although the incisions in the SHS technique may be larger than those in IMNs, thorough hemostasis can be achieved in the former, which may reduce the incidence of postoperative hematoma, tumor seeding, and infection. Furthermore, the costs of the SHS construct are much lower than those of the locking plate, IMN, and arthroplasty, especially when taking into consideration the heavy financial burden on terminal patients and their families.

Possible complications of the conventional SHS technique include lag screw cut-out, avascular necrosis of the femoral head, cortical screw pullout or fracture, and side-plate or barrel fracture due to a lack of load-sharing between the implant and residual bone [14,15]. While treating pathologically impending or actual peritrochanteric fractures, cortical screw failures are more common after cement augmentation. A possible reason is that cement augmentation prevents the lagging function in the SHS bolt by making the peritrochanteric bone void more rigid and by improving the lag screw purchase. Mechanical loading is shared between the compressed bone and lag screw in conventional SHS application, and the shearing force of the cortical screws in the side-plate portion of the conventional SHS with cement augmentation increases while the rigid lag screw acts as the fulcrum of the lever (Figure 6a). Thus, in cases of delayed union or non-union, the cortical screws of the side plate became the weak points in the construct (Figure 6b). We attempted to design a novel technique to overcome such mechanical failures by integrating a cerclage reconstruction plate with the distal part of the side plate and the femoral shaft to increase the strength of the construct against the shearing force (Figure 6c).

In our study, regardless of which method was used to estimate blood loss, augmentation with a cerclage reconstruction plate did not significantly affect the intraoperative or perioperative blood loss. We routinely used TXA both preoperatively (intravenous infusion) and postoperatively (topical). According to a prospective randomized controlled trial conducted by Tian et al., patients with femoral intertrochanteric fractures treated with IMNs had less total blood loss in the TXA group than in the control group (515.30 ± 278.79 mL versus 696.88 ± 275.00 mL; *p* = 0.005), which was close to our results (group A, 511.4 ± 366 mL; group B, 719 ± 406 mL) [16]. Interestingly, we observed a trend toward lower blood loss in the augmentation group; this was difficult to explain since the surgical dissections were more extensive, and the surgery durations were longer in the augmentation group. A possible explanation for this is that all methods that were used to estimate the perioperative blood loss did not consider intravenous and oral hydration, which might have had some influence on the postoperative hemoglobin and hematocrit. Moreover, our sample size was not large enough to generate more conclusive results. The surgery duration was the only variable in this study that was significantly different between the groups. Augmentation with a cerclage reconstruction plate around the side plate and femoral shaft increased the mean surgery duration by 24.6 min.

This study has several limitations. First, the retrospective, non-randomized design and small sample size made it difficult to analyze the true effect of cerclage augmentation. Second, we only identified implant fractures if the patient came back to our clinic and was evaluated using plain film radiographs. Some patients might go to other hospitals for treatment after an implant failure. Third, we usually perform the modified SHS technique in revision surgeries or more difficult cases, which might also be the reason behind the non-superior results obtained with the augmented construct compared with conventional SHS.

## 5. Conclusions

To the best of our knowledge, this is the first study to report on the use of the same construct. SHS constructs augmented with cerclage reconstruction plates were associated with a longer surgery duration, though not with increased blood loss. The competing risk analysis showed no significant difference in implant survival between the two groups; however, all cases that failed in group B and revised with modified SHS were successful and the implant survival time was >2 years than that observed with the previous conventional SHS constructs. Our modified SHS technique may provide an alternative solution for the treatment of challenging pathologic (and impending) peritrochanteric fractures and serves as a procedure to salvage cases with a history of failed SHS constructs because patients with cancer have recently achieved longer life expectancies.

## Figures and Tables

**Figure 1 jcm-10-04271-f001:**
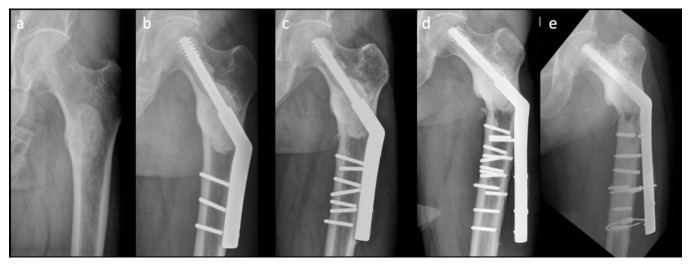
A difficult case with multiple implant failures: (**a**) preoperative X-ray image, (**b**) first-time implant failure, (**c**) second-time implant failure, (**d**) third-time implant failure, and (**e**) fourth-time implant failure.

**Figure 2 jcm-10-04271-f002:**
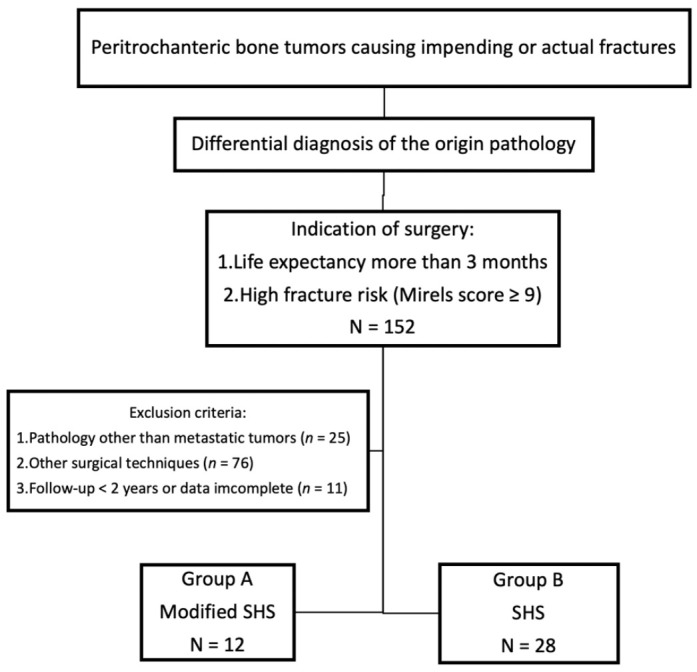
Flowchart of the study.

**Figure 3 jcm-10-04271-f003:**
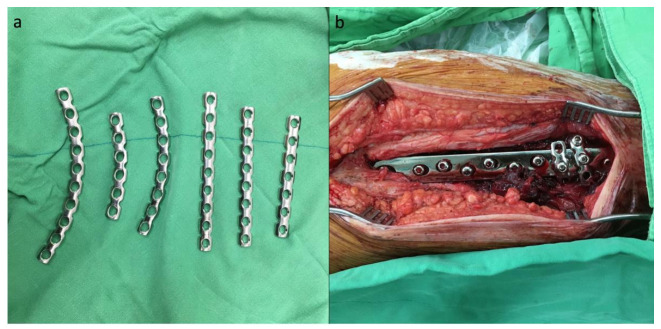
Surgical procedure of modified SHS: (**a**) a 3.5 mm AO reconstruction plate was chosen after consideration of the curve and length; (**b**) intraoperative photograph of the cerclage reconstruction plate.

**Figure 4 jcm-10-04271-f004:**
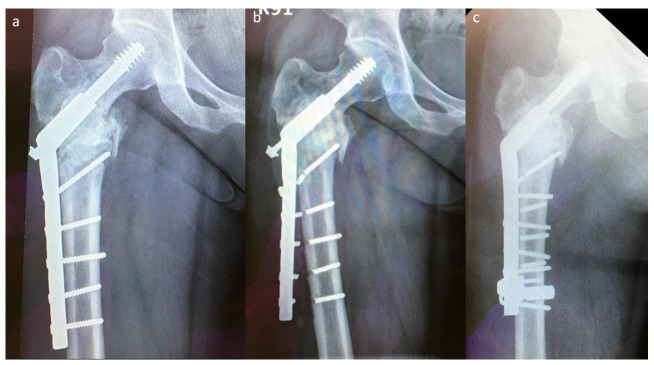
An implant failure case in group B: (**a**) original postoperative X-ray image, (**b**) first implant failure, and (**c**) postoperative plain radiograph after revision surgery with modified SHS.

**Figure 5 jcm-10-04271-f005:**
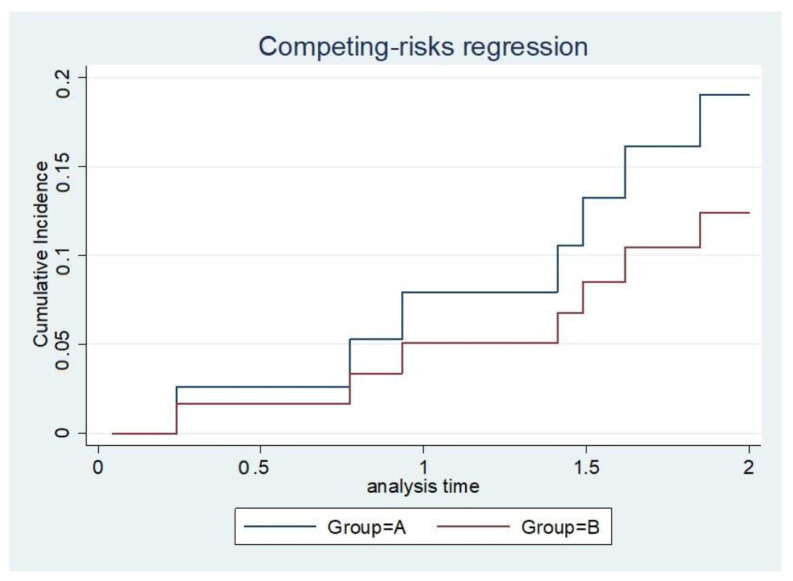
Competing risk regression. Analysis time (year).

**Figure 6 jcm-10-04271-f006:**
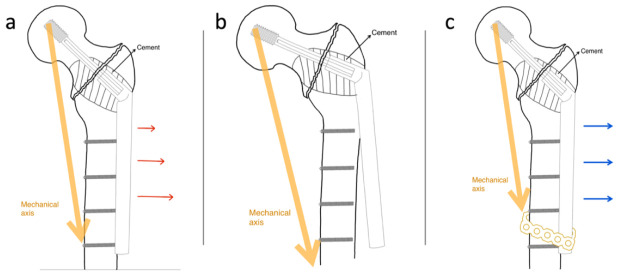
Illustration of the change in shearing force: (**a**) the shearing force increased due to PMMA cement augmentation, (**b**) loading failure with distal cortical screw fracture with SHS fixation only, and (**c**) the modified SHS technique increases the strength to resist the shearing force.

**Table 1 jcm-10-04271-t001:** Patient demographic data.

		Group A *n* = 12	Group B *n* = 28	*p*-Value
Age (years)		67.0 (62.5, 75)	64.0 (55.5, 72.5)	0.375
Sex	Male/Female	8/4 (66.7%/33.3%)	12/16 (42.9%/57.1%)	0.301
Body weight (kg)		59.4 (53.1, 63.7)	57 (49.7, 65.8)	0.479
Survival (days)		381.5 (159.5, 730)	290.5 (128.5, 495)	0.523
Surgery duration (min)		125 (99, 149.5)	93 (83, 105)	0.013 *
Preoperative lesion status	Impending fracture	3 (25%)	18 (64.3%)	0.038 *
	Actual fracture	9 (75%)	10 (35.7%)	
Tumor type	Lung cancer	4	7	
	Breast cancer	1	5	
	Hepatocellular carcinoma	0	3	
	Renal cell carcinoma	2	1	
	Multiple myeloma	1	2	
	Pleomorphic sarcoma	0	2	
	Prostate cancer	0	2	
	Nasopharyngeal cancer	0	1	
	Bladder cancer	1	0	
	Squamous cell carcinoma	1	0	
	Cholangiocarcinoma	1	0	
	Undifferentiated carcinoma	0	1	
	Chondrosarcoma	0	1	
	Endometrial cancer	0	1	
	Fibrous dysplasia	1	0	
	Intraosseous myelolipoma	0	1	
	Fibrosis and calcification	0	1	

* Significant differences between the two groups.

**Table 2 jcm-10-04271-t002:** Complications in groups A and B.

	Group A (*n* = 12)	Group B (*n* = 28)
Wound complication	1	2
Tumor progression or recurrence	1	3
Implant failure requiring revision surgery	2	5

**Table 3 jcm-10-04271-t003:** Factors associated with implant failure and revision surgery.

	Unadjusted HR (95% CI)	*p*-Value ^a^
Group		
A	0.59 (0.11, 3.24)	0.549
B	ref.	
Tumor location		
No	ref.	
Yes	2.95 (1.29, 6.75)	<0001 *
Postoperative VAS	2.19 (1.35, 3.57)	0.002 *
Radiotherapy	5.33 (0.94, 30.39)	0.059

Abbreviations: HR, hazard ratio; VAS, visual analog scale. Tumor location: Yes, with involvement of the subtrochanteric region; No, without involvement of the subtrochanteric region. ^a^ *p*-values are derived from the Cox regression model including all variables in the table. * Significant differences between the two groups.

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
