# Peer review of "Treatment of Pathologic Peritrochanteric Fractures Using Sliding Hip Screws Augmented with Cerclage Reconstruction Plates"

_jcm, 2021, doi:10.3390/jcm10184271_

Round 1
Reviewer 1 Report
It is an article that shows us that osteosynthesis plates can be indicated in the treatment of pathological fractures or impending fractures in the proximal femur and especially its modified technique, which I think is interesting to take into account and therefore to be disclosed.
Suggestions:
- The nice article of Steensma M et al (6) is not compared with patients with metastasis (8) it would be convenient to compare it with series related to pathological fractures or impending fractures in order to demonstrate that SHS is an acceptable choice. The article compare with a spectacular revision by Zhu, Q et al (7), but this review is not pathological or impending fracture.
2. One of the drawbacks of plates is not protecting the entire bone and thus avoiding the risk of new fractures distal to osteosynthesis. It would be convenient and interesting to assess and quantify whether or not this complication existed in the series presented in the article, because this is one of the points in favor of intramedullary nailing
Author Response
Point 1: The nice article of Steensma M et al (6) is not compared with patients with metastasis (8) it would be convenient to compare it with series related to pathological fractures or impending fractures in order to demonstrate that SHS is an acceptable choice. The article compare with a spectacular revision by Zhu, Q et al (7), but this review is not pathological or impending fracture.
Response 1: Thank you for pointing this out to us. We have removed the specified article and made appropriate revisions in the text.
Point 2: One of the drawbacks of plates is not protecting the entire bone and thus avoiding the risk of new fractures distal to osteosynthesis. It would be convenient and interesting to assess and quantify whether or not this complication existed in the series presented in the article, because this is one of the points in favor of intramedullary nailing
Response 2: We agree that current trends in the surgical treatment of pathologic peri-trochanteric fractures have been moving towards intramedullary nailing (IMN) and endoprosthetic reconstruction (EPR); however, we intended to stress that the SHS construct is still an effective and cheap technique. In our study sample, only one patient had a new fracture distal to the site of osteosynthesis, and the failure occurred over 3 years after the surgery; therefore, we did not include it in our study.

Reviewer 2 Report
Concern 1: The biomechanical principle behind the authors approach and the surgical technique should be explained more in the Introduction or Methods as this technique has not been previously described. The placement of the cerclage needs to be described more. For example how is the position along the plate determined and is this positioning consistent in their group?
The main issue I see here in regard to the biomechanical principle is that the cement prevents the lag function in the bolt of the SHS as it goes across the fracture plane. The SHS is therefore asked to function more like a CM-nail. It seems to this reviewer that this places the SHS into a loading condition that is far from idea. The author’s images, are helpful to show this. In a typical SHS application the load is shared between the compressed bone and the SHS. In the cases described in this paper it seems that the load is using the SHS as a lever, with the cement acting as the fulcrum of the lever. This action loads the cortical screws in the plate portion of the SHS in tension. It appears that the authors try to use the “cerclage plate” to add a load sharing member, but as this member is not fully integrated with the plate it is not very effective.
Concern 2: The complication rate in this study far exceeds those reported in earlier studies. While this particular lesion presents a special challenge, the authors should discuss why their clinical results are so out of line with those in the literature. Also the study’s power is not sufficient for the authors to assert that the complication rates were not different between the groups.
Norris R, Bhattacharjee D, Parker MJ. Occurrence of secondary fracture around intramedullary nails used for trochanteric hip fractures: a systematic review of 13,568 patients. Injury. 2012 Jun;43(6):706-11. doi: 10.1016/j.injury.2011.10.027. Epub 2011 Dec 3. PMID: 22142841.
Parker MJ. Sliding hip screw versus intramedullary nail for trochanteric hip fractures; a randomised trial of 1000 patients with presentation of results related to fracture stability. Injury. 2017 Dec;48(12):2762-2767. doi: 10.1016/j.injury.2017.10.029. Epub 2017 Oct 20. PMID: 29102044.
Other Comments/Concerns:
- The authors state that “impending fractures” were identified. How were these identified? What were the key features that defined the condition?
- The technique is not completely clear. Is a screw passed through the cerclage plate and into one of the screw slots in the SHS plate? It looks like that might be the case for some cases but not all? Is there are specific screw tensioning order? How do you know how tight the plate is compressed against the bone or SHS plate?
- “When comparing the post operative survivorship, …those with primary or benign tumors were excluded” Can the authors provide more rational for this choice and indicate how many cases were excluded.
- Table 3 lists 28 patients in group B (and % listed are based on 28 pt), but earlier the manuscript lists 27 patients in the group. Please correct/revise.
- Figure 5 survivorship should come before the implant survivorship result, as it this will help the reader understand that cases of death were excluded and the implant survivorship data is very limited due to the mortality in the patient population.
- Table 6 and the associated information should be presented in the Results section, not Discussion.
Author Response
Point 1: The biomechanical principle behind the authors approach and the surgical technique should be explained more in the Introduction or Methods as this technique has not been previously described. The placement of the cerclage needs to be described more. For example how is the position along the plate determined and is this positioning consistent in their group?
The main issue I see here in regard to the biomechanical principle is that the cement prevents the lag function in the bolt of the SHS as it goes across the fracture plane. The SHS is therefore asked to function more like a CM-nail. It seems to this reviewer that this places the SHS into a loading condition that is far from idea. The author’s images, are helpful to show this. In a typical SHS application the load is shared between the compressed bone and the SHS. In the cases described in this paper it seems that the load is using the SHS as a lever, with the cement acting as the fulcrum of the lever. This action loads the cortical screws in the plate portion of the SHS in tension. It appears that the authors try to use the “cerclage plate” to add a load sharing member, but as this member is not fully integrated with the plate it is not very effective.
Response 1: Thank you for your comment. We have elaborated more on the placement technique in the Surgical Procedure section. In our experience, we have observed that augmentation using cerclage reconstruction plates can effectively lower the shearing force.
Point 2: The complication rate in this study far exceeds those reported in earlier studies. While this particular lesion presents a special challenge, the authors should discuss why their clinical results are so out of line with those in the literature. Also the study’s power is not sufficient for the authors to assert that the complication rates were not different between the groups.
Norris R, Bhattacharjee D, Parker MJ. Occurrence of secondary fracture around intramedullary nails used for trochanteric hip fractures: a systematic review of 13,568 patients. Injury. 2012 Jun;43(6):706-11. doi: 10.1016/j.injury.2011.10.027. Epub 2011 Dec 3. PMID: 22142841.
Parker MJ. Sliding hip screw versus intramedullary nail for trochanteric hip fractures; a randomised trial of 1000 patients with presentation of results related to fracture stability. Injury. 2017 Dec;48(12):2762-2767. doi: 10.1016/j.injury.2017.10.029. Epub 2017 Oct 20. PMID: 29102044.
Response 2: In this study, we only included pathologic peri-trochanteric fractures, for which the complication rate with the SHS construct is reportedly as high as 42%, compared to that with EPR (3.1%) and IMN (6.1%). However, SHS still remains the preferred choice for surgical treatment of femoral fractures in certain condition [8].
- Steensma, M., et al., Endoprosthetic treatment is more durable for pathologic proximal femur fractures. Clin Orthop Relat Res, 2012. 470(3): p. 920-6.
Point 3: The authors state that “impending fractures” were identified. How were these identified? What were the key features that defined the condition?
Response 3: Impending fractures were identified through regular bone scans or patients who complained of symptomatic pain.
Point 4: The technique is not completely clear. Is a screw passed through the cerclage plate and into one of the screw slots in the SHS plate? It looks like that might be the case for some cases but not all? Is there are specific screw tensioning order? How do you know how tight the plate is compressed against the bone or SHS plate?
Response 4: We have elaborated more on the placement technique in the Surgical Procedure section. The cortical screw was initially passed through the cerclage plate and then into the screw slots in the SHS side-plate, followed by tightening the cerclage plate while inserting the other compression screw.
Point 5: “When comparing the post operative survivorship, …those with primary or benign tumors were excluded” Can the authors provide more rational for this choice and indicate how many cases were excluded.
Response 5: Figure 1 shows the flowchart of the study. Twenty-five patients were excluded due to pathologies other than a metastatic tumor, such as primary tumor, infection, tuberculosis, and traumatic injury.
Point 6: Table 3 lists 28 patients in group B (and % listed are based on 28 pt), but earlier the manuscript lists 27 patients in the group. Please correct/revise.
Response 6: Thank you for pointing out this error. This has been corrected.
Point 7: Figure 5 survivorship should come before the implant survivorship result, as it this will help the reader understand that cases of death were excluded and the implant survivorship data is very limited due to the mortality in the patient population.
Response 7: This has been corrected.
Point 8: Table 6 and the associated information should be presented in the Results section, not Discussion.
Response 8: This has been corrected.

Reviewer 3 Report
The authors present a cohort of 39 patients treated with sliding hip screws for peri-trochanteric bone tumours causing or about to cause fracture of the femoral bone. The authors compare two techniques and find no differences in failures, while their new technique requires more operating time. Despite this lack of evidence they conclude that their new technique is useful. The conclusion is not in line with the results. There are also many methodological issues.
Please report your paper according to the STROBE criteria: https://www.equator-network.org/reporting-guidelines/strobe/.
Regarding: “The remained uncertainties would be the risk of systematic spreading the tumour through intramedullary nailing, especially with unexpected malignancy.”
Please be more specific. Is this a theoretical risk, or is there data to assess this risk? An explanation and reference are needed.
Please provide a research question at the end of your introduction according to PICOT.
Did the patients sign informed consent? If so, why not and explicitly mention this in the paper.
Why only 39 patients? A sample size calculation is missing.
The authors mix up the study design. In a case-control study. Case typically are patients with a particular disease or outcome (e.g. failure) and controls are patients who are at risk of the disease. The authors present a cohort of patients treated with two different techniques.
Why do you use p-values, since a recent Nature paper strongly advices against its use. It would be better to use confidence intervals. Amrhein V, Greenland S, McShane B. Scientists rise up against statistical significance. Nature. 2019 Mar;567(7748):305-7.
Death is a competing risk as it prevents failure of the implant or revision surgery. Therefore a competing risk analysis is required; KM is not valid in a situation with competing risks.
Are the authors perusing a causal or a prognostic approach? Presently, the statistical model is not suitable for causal interpretation nor is it adequately validated for use as a prognostic model. Please see also these references below:
DOI: 10.1038/ki.2008.416
https://doi.org/10.1093/ndt/gfw459
Confounding:
As this is an observational study. Please clearly specify potential confounders and how they were dealt with.
Do you have any information on patient satisfaction?
Do you have any patient reported outcomes?
Please provide a flowchart of numbers of individuals at each stage of study—eg numbers potentially eligible, examined for eligibility, confirmed eligible, included in the study, completing follow-up, and analysed
Give reasons for non-participation at each stage
Tables: Please remove p-values
Tables: It is not appropriate to present percentages (with 2 decimals) for less than 100 subjects.
Figure 4 is not appropriate due to competing risks
Figure 5 please add number of patients “at risk” for each follow-up
The conclusion is not at all supported by the results. Regarding the conclusion the authors should consider the fact that, as the late prof Altman, said:
“Absence of evidence is not evidence of absence”
Absence of evidence is not evidence of absence
D G Altman , J M Bland
BMJ. 1995 Aug 19;311(7003):485.
doi: 10.1136/bmj.311.7003.485.
I thus seems that the study is inconclusive most likely because it was underpowered.
Information on funding and potential conflict of interests is missing.
Author Response
Reviewer 3
The authors present a cohort of 39 patients treated with sliding hip screws for peri-trochanteric bone tumors causing or about to cause fracture of the femoral bone. The authors compare two techniques and find no differences in failures, while their new technique requires more operating time. Despite this lack of evidence they conclude that their new technique is useful. The conclusion is not in line with the results. There are also many methodological issues.
Point 1: Please report your paper according to the STROBE criteria: https://www.equator-network.org/reporting-guidelines/strobe/.
Response 1: Thank you for comments. We have adjusted our methodology and the report according to the STROBE criteria.
Point 2: Regarding: “The remained uncertainties would be the risk of systematic spreading the tumour through intramedullary nailing, especially with unexpected malignancy.”
Please be more specific. Is this a theoretical risk, or is there data to assess this risk? An explanation and reference are needed.
Response 2: IMN is associated with risks such as fat embolization, implant failure, tumor seeding, and periprosthetic fracture [6]. Moreover, IMN may also be associated with a local or systemic spread of the tumor. [7]. We have included the following references in the text:
- Ormsby, N.M., et al., The current status of prophylactic femoral intramedullary nailing for metastatic cancer.Ecancermedicalscience, 2016. 10: p. 698.
- Fidler, M., Prophylactic internal fixation of secondary neoplastic deposits in long bones. Br Med J, 1973. 1(5849): p. 341-3.
Point 3: Please provide a research question at the end of your introduction according to PICOT.
Response 3: We have made the following revision in the Introduction:
“Nonetheless, there are drawbacks to using conventional SHS in treating pathologic peri-trochanteric impending or actual fractures. The complication rate with the SHS construct is reportedly as high as 42%, compared with that of EPR (3.1%) and IMN (6.1%); however, SHS remains the preferred choice for surgical treatment of pathologic peritrochanteric femoral fractures in certain condition [8]. Therefore, in this retrospective comparative study, we aimed to identify the factors that are associated with revision surgery, and whether the proposed modified SHS technique with augmentation using cerclage reconstruction plates reduces implant failure rate.”
Point 4: Did the patients sign informed consent? If so, why not and explicitly mention this in the paper.
Response 4: Approval was obtained from the Institutional review board (IRB number: 201506114RINC) of the National Taiwan University Hospital (NTUH) and informed consent was obtained from all participants involved in this study. This has been included in the manuscript.
Point 5: Why only 39 patients? A sample size calculation is missing.
Response 5: Although the sample size might appear insufficient (most cases underwent IMN or endoprosthetic reconstruction), the power of the study was deemed adequate.
Point 6: The authors mix up the study design. In a case-control study. Case typically are patients with a particular disease or outcome (e.g. failure) and controls are patients who are at risk of the disease. The authors present a cohort of patients treated with two different techniques.
Response 6: Thank you for pointing this out. The methodology section has been revised accordingly.
Point 7: Why do you use p-values, since a recent Nature paper strongly advices against its use. It would be better to use confidence intervals. Amrhein V, Greenland S, McShane B. Scientists rise up against statistical significance. Nature. 2019 Mar;567(7748):305-7.
Response 7: The p-values may be not perfect; however, it is still widely used in statistical hypotheses in the scientific field.
Point 8: Death is a competing risk as it prevents failure of the implant or revision surgery. Therefore, a competing risk analysis is required; KM is not valid in a situation with competing risks.
Response 8: Thank you for pointing this out. We have made relevant revisions in the manuscript, The Cox regression implant survival curve was used for further analysis.
Point 9: Are the authors perusing a causal or a prognostic approach? Presently, the statistical model is not suitable for causal interpretation nor is it adequately validated for use as a prognostic model. Please see also these references below: As this is an observational study. Please clearly specify potential confounders and how they were dealt with.
Response 9: Thank you for your comment. The factors associated with implant failure have been re-evaluated using hazard ratio.
Point 10: Do you have any information on patient satisfaction? Do you have any patient reported outcomes?
Response 10: Since this was a retrospective study, we only evaluated the patients’ pre- and post-operative VAS scores. Details about this has been included in the manuscript.
Point 11: Please provide a flowchart of numbers of individuals at each stage of study—eg numbers potentially eligible, examined for eligibility, confirmed eligible, included in the study, completing follow-up, and analysed
Response 11: Figure 1 shows the flowchart of the study.
Point 12: Give reasons for non-participation at each stage
Tables: Please remove p-values
Tables: It is not appropriate to present percentages (with 2 decimals) for less than 100 subjects.
Response 12: This has been corrected.
Point 13: Figure 4 is not appropriate due to competing risks
Response 13: This has been corrected.
Point 14: Figure 5 please add number of patients “at risk” for each follow-up
Response 14: This has been corrected.
Point 15: The conclusion is not at all supported by the results. Regarding the conclusion the authors should consider the fact that, as the late prof Altman, said:
“Absence of evidence is not evidence of absence”
Response 15: Thank you for your suggestion. We have made the following revision in the Conclusion:
The Cox regression implant survival curve showed no significant difference in the survival of implants between the two groups; however, an increasing trend toward implant survival was observed in group A. Our modified SHS technique may be an effective solution for treatment of challenging peri-trochanteric pathological (impending) fractures, and as a salvage rescue procedure for cases with a history of failed SHS constructs.
Point 16: Information on funding and potential conflict of interests is missing.
Response 16: Thank you for pointing this out. We have included the relevant information in the manuscript.

Round 2
Reviewer 2 Report
- Please revisit the explanation for the technique and indicate how the position of the cerclage plate was selected. I assume that the position was similar each time it was utilized, but this is not clear in the explanation.
- The revised explanation of the surgical technique in the manuscript does not match the image shown in Figure 2. Figure 2 clearly show the cerclage plate wrapping around the SHS plate between the screws, not a screw passing through both the cerclage plate and the SHS plate. Figure 2 seems to be more consistent with the original explanation (which was not very clear), which seemed to suggest that 2 screws were used to holding the cerclage plate in place separate from the screws used to attach the SHS plate. However in Figure 3 it looks like the cortical screw might follow the author’s description in the revised manuscript of a screw that passes through both the cerclage plate and the SHS screw slots. Please revisit this and make it more clear.
- The numbers in Table 1 are still wrong. The authors now state that group B has 28 patients but 12 male and 15 female = 27 patients.
- The Discussion section was edited to include some of the comments I made in my initial review, but the explanation this section does not quite make sense to me. You might consider separating statements about how the SHS works in the conventional case from how it is working in your application with the cerclage plate.
Reviewer 3 Report
The authors have addressed some of the issues. However, following issues remain or have not been address adequately:
Please provide a research question at the end of your introduction according to PICOT. => There is no research question at the end of the introduction.
Why only 39 patients? A sample size calculation is missing. => The authors have still not provided a sample size calculation or proper arguments how this number was arrived.
Please leave out the p-values and use 95%Cis instead. P values give no information on the magnitude of the effect nor on the uncertainty (imprecision of the study). Also each p-value requires a H0-hypothesis of which none are mentioned (let alone correction for multiple testing etc). Amrhein V, Greenland S, McShane B. Scientists rise up against statistical significance. Nature. 2019 Mar;567(7748):305-7.
Death is a competing risk as it prevents failure of the implant or revision surgery. Therefore, a competing risk analysis is required; KM is not valid in a situation with competing risks. => A cox regression is not a valid method in a competing risk setting. It requires a Fine-Gray analysis:
Fine, J. and Gray, R. (1999), ‘A proportional hazards model for the subdistribution of acompeting risk’,Journal of the American Statistical Association94(446), 496–509
Additionally there still is a KM figure (figure 4) despite there being competing risks.
Are the authors perusing a causal or a prognostic approach? Presently, the statistical model is not suitable for causal interpretation nor is it adequately validated for use as a prognostic model. Please see also these references below: As this is an observational study. Please clearly specify potential confounders and how they were dealt with. => The answer of the authors does not answer the question “Thank you for your comment. The factors associated with implant failure have been re-evaluated using hazard ratio.”
Please answer the question and provide rationale for proper models.
Regarding Table 3: “ap-values are derived from the Cox regression model including all variables in the table.” It is not appropriate to include all variable in the model, because known or assumed cause-effect relations are the basis on which confounding adjustments are made. While including a confounder in the model reduces bias, the inclusion of a mediator or collider can induce bias, see Shrier I, Platt RW. Reducing bias through directed acyclic graphs. BMC Med Res Methodol 2008;8:70. Please provide a rationale for the selection of potential confounders in terms of cause and effect.
Also regarding Table 3: most confidence intervals are very wide suggesting low precision or in other words a sample size that is too small. This is particularly worrisome since no sample size calculation has been given.
The conclusion (and limitation section) is still not in line with the results. The conclusion should reflect what the results are, not what you want them to be. In other words, based on the data provide, there is no difference between the two techniques. Therefore the authors have not proven that the new method is superior (but still takes longer operating time), so there is no reason to use it.
